psychology

intergroup contact, mentalizing, motivation, theory of mind

**Author for correspondence:**
Jasmin Cloutier
e-mail: jclout@udel.edu

†Authors contributed equally to the work.

# Impact of interracial contact on inferring mental states from facial expressions

Grace Handley[1,†], Jennifer T. Kubota[1,2,†], Tianyi Li[3] and Jasmin Cloutier[1]

[1]Department of Psychological and Brain Sciences, and [2]Department of Political Science and International Relations, University of Delaware, DE, USA
[3]College of Business Administration, University of Illinois Chicago, IL, USA

GH, 0000-0002-5170-0563; JTK, 0000-0002-0366-8490

Although decades of research have shown that intergroup contact critically impacts person perception and evaluation, little is known about how contact shapes the ability to infer others' mental states from facial cues (commonly referred to as mentalizing). In a pair of studies, we demonstrated that interracial contact and motivation to attend to faces jointly influence White perceivers' ability to infer mental states based on facial expressions displaying secondary emotions from both White targets alone (study 1) and White and Black targets (study 2; pre-registered). Consistent with previous work on the effect of motivation and interracial contact on other-race face memory, we found that motivation and interracial contact interacted to shape perceivers' accuracy at inferring mental states from secondary emotions. When motivated to attend to the task, high-contact White perceivers were more accurate at inferring both Black and White targets' mental states; unexpectedly, the opposite was true for low-contact perceivers. Importantly, the target race did not interact with interracial contact, suggesting that contact is associated with *general* changes in mentalizing irrespective of target race. These findings expand the theoretical understanding and implications of contact for fundamental social cognition.

## 1. General introduction

For over 60 years, scholars have highlighted how intergroup contact fundamentally shapes how we think and feel about others [1]. Contact critically impacts intergroup relations, reducing both explicit and implicit racial biases [2]. However, few studies have considered how interracial contact may more broadly shape social cognition in general and peoples' ability to infer others' mental states more specifically. Recent brain imaging research hints at a possible relationship between interracial contact and people's

ability to infer mental states from facial expressions displaying secondary emotions, observing that increased interracial contact impacts the recruitment of brain networks involved in face processing, salience detection and social cognition [3]. Following these findings, the present research is, to our knowledge, the first to explore the social cognitive consequences of interracial contact, focusing on how contact and motivation to attend to others shape White perceivers' ability to infer mental states from secondary emotions displayed by both same- (White) and cross-race (Black) faces.

Given that little work to date has tested how interracial contact specifically impacts mental state inferences about others, we present two studies examining how interracial contact shapes White perceivers' ability to infer mental states from only same-race targets (i.e. White individuals; Study 1) and from same-race and cross-race targets (i.e. White and Black individuals; Study 2). Although interracial contact has typically been studied in the context of how it impacts the perception and evaluation of cross-race targets—work which we highlight later—previous neuroimaging research showing that increased contact shapes the recruitment of brain networks involved in face processing and social cognition *regardless of target race* [3] led us to expect that contact may also impact mental state inferences from same-race targets. Thus, Study 1 tested only same-race targets. In Study 2, we extended this work to include both same-race and cross-race targets based on the extensive body of work suggesting contact shapes how cross-race targets are perceived.

Inferring others' mental states and understanding the distinctness of their minds are fundamental aspects of social cognition necessary to successfully navigate social environments. Theory of mind and the processes involved in mentalizing about others have been studied extensively within the context of developmental psychology [4–6]; however, mentalizing about outgroup members among both children and adult populations has been minimally studied [7–9]. There is therefore a major gap in the literature regarding the impact of individual differences, such as differences in interracial contact, on how people infer others' mental states across social categories like race, particularly within the social context of the United States. In the present work, we investigated how White Americans' history of interracial contact affects their performance on the Reading the Mind in the Eyes (RME) task, a task that assesses how accurately people can infer mental states from photographs of others' eye region of the face displaying secondary emotions [10,11]. We specifically used two versions of the RME task: one version that uses exclusively White stimuli [10] and one version that uses these same White stimuli *and* Black stimuli that expressed the same target emotions and were equated on difficulty with the White stimuli (see [11]). Although this task is not well characterized in neurotypical populations [12], we chose to use it for a number of reasons. First, it is possible to visually manipulate the target race using the RME task because it is a photograph-based task. Although the original task uses exclusively White stimuli, we developed and equated a matched task using Black stimuli to facilitate the present research [11]. Second, the RME task is one of the few visual tasks that can assess adults' mentalizing *accuracy* rather than mentalizing speed; whereas neurotypical adults tend to perform at the ceiling in other established mentalizing tasks (e.g. Sally-Anne task; [13]). Finally, the RME has previously been used successfully in other studies of adult social cognition [14,15].

We specifically aimed to test the impact of peoples' history of interracial contact on their ability to infer mental states from displays of secondary emotions following previous work suggesting that contact, as a form of expertise with other-race individuals, influences a broad range of person perception processes other than mentalizing. For example, contact notably influences people's attitudes toward outgroup members, particularly implicit attitudes, with higher levels of contact typically being associated with either more positive outgroup implicit attitudes or less positive ingroup implicit attitudes [16–19]. In addition to these changes in attitudes, contact is associated with changes in social categorization. For example, increased contact with mixed-race individuals is associated with decreased race essentialism [20,21], and low-contact individuals are less efficient at categorizing mixed-race targets as either 'Black' or 'White' than high-contact individuals [22]. Contact also shapes face attention and memory, as demonstrated by research on how varying levels of interracial exposure affect young infants' preferential looking times at different race faces. Preference for own-race faces emerges in infants as early as three months old; however, this preference is largely driven by increased exposure to own-race faces and is reduced among infants with greater cross-race face exposure [23–28]. Given the breadth of social cognitive processes that are influenced by peoples' cross-race experiences, it would not be surprising that contact would also critically shape our ability to infer others' mental states.

Brain imaging evidence further highlights the importance of intergroup contact in shaping person perception and evaluation. Investigations exploring how the amygdala, a brain region involved in processing salient emotional stimuli, responds to racial outgroup members have shown that interracial

contact modulates amygdala responses to outgroup faces [29,30]. Additional work suggests that contact exerts an important top-down influence on social vision, particularly the neural substrates of social categorization and race perception [31,32]. Furthermore, contact may reduce the racial ingroup bias in empathy, as evidenced by a positive association between contact and anterior cingulate cortex activity when viewing outgroup members in pain (for review, see [33]). This research highlights how contact fundamentally shapes intergroup processes even at the neural level, but this line of research has yet to consider how contact impacts mental state inferences about others.

Although little research has investigated how contact may influence peoples' inferences of others mental states, related work on contact and outgroup dehumanization lends some insight into how contact may affect how people understand others' emotions and mental states (for review, see [34]). For example, high-quality contact with various outgroups was associated with decreased blatant animalistic dehumanization toward members of those groups and decreased expectations of being dehumanized themselves by those outgroup members [35]. This work shows the promising positive effects of contact on explicitly humanizing outgroup members and prioritizing their perspectives, which might lead to the prediction that high contact would be associated with more accurate mentalizing, particularly when failing to do so may lead to discrimination. However, recent neuroimaging research raises an alternate possibility. Contact, as a form of expertise with a wide variety of faces, may increase face processing efficiency (i.e. higher expertise leads to more efficient processing; see [36–38]). In contrast with the dehumanization literature, in which perceivers make explicit, intentional and potentially discriminatory judgements about outgroup members, a study found that in more mundane everyday situations high-contact perceivers may process faces efficiently and with little concerted effort, leading to worse overall mentalizing in such scenarios [3]. More specifically, high-contact White perceivers showed decreased recruitment of a network of brain regions involved in face processing, salience detection and social cognition (including regions associated with mentalizing) when forming passive impressions of Black and White faces; these changes did not differ as a function of target race [3]. However, in light of the suggestions that increased contact reduces dehumanization (for review, see [34]), this finding does not necessarily suggest that high-contact perceivers are always worse mentalizers. Instead, high-contact perceivers may further engage in mentalizing about others only when the *motivation* to do so is sufficient.

In line with the possibility that high-contact perceivers may require motivation to overcome relatively lower spontaneous social cognitive engagement, Young & Hugenberg [39] found that high-contact perceivers who were motivated to attend to faces had better recognition for other-race faces, whereas motivation did not improve other-race face recognition among low-contact perceivers [39]. Expert (high contact) face processors may uniquely benefit from being motivated to attend to faces because their baseline cognitive engagement with faces is relatively low [3,39]. Although Young & Hugenberg's [39] research suggests that motivation may modulate the relationship between contact and intergroup face recognition, it does not address how contact shapes mentalizing. The present work therefore aims to examine the novel question of how contact and motivation impact mentalizing performance.

## 1.1. Study overview

In a pair of studies, we explored whether White perceivers' history of interracial contact was associated with changes in mentalizing from same-race (i.e. White; study 1) and same- and cross-race (i.e. White and Black; study 2) individuals using the RME test [10]. Because the RME is not an especially salient or attention-grabbing task, we posited that high-contact perceivers may not spontaneously infer mental states from newly encountered same-race *and* cross-race faces due to increased face processing efficiency at the expense of diminished everyday social cognitive engagement [3]. However, this impairment may be overcome by perceivers who are motivated to more deeply process faces [39]. These predictions were explicitly pre-registered for study 2 (https://osf.io/v8wgg/).

# 2. Study 1

## 2.1. Introduction

In study 1, we explored how interracial contact influences mentalizing from same-race faces (i.e. White perceivers inferring mental states from only White RME targets). Because motivation to deeply process faces may enable high-contact perceivers to overcome decreased social cognitive engagement resulting from increased face processing efficiency, we manipulated motivation to attend to targets in a

between-subjects manner. We compared performance on the RME test with only White targets among White American perceivers varying in interracial contact.

### 2.1.1. Hypotheses

Because face processing efficiency may be increased among high-contact perceivers [3], we predicted that high-contact perceivers would be less accurate at RME than low-contact perceivers when they were not motivated to deeply process faces, reflecting diminished baseline social cognitive engagement. However, due to high-contact perceivers' increased overall expertise with a broad range of faces, we predicted that these perceivers could overcome their decreased social cognitive engagement when sufficiently motivated [39] and be more accurate than low-contact perceivers at RME. We therefore hypothesized that the interaction between interracial contact and perceiver motivation would significantly predict same-race RME accuracy.

## 2.2. Method

### 2.2.1. Participants

All US-based workers from Amazon Mechanical Turk (MTurk) were eligible to participate. We recruited 175 participants ($M_{age} = 32.19$, s.d. = 7.85, 91 female) through MTurk who were paid \$1.50 for their participation. Two participants completed the RME test twice. Only their first completion was included in analyses. All participants identified as White, were between the ages of 18 and 50 years old, and were born in the United States.

### 2.2.2. Power analysis

We ran a *post hoc* power analysis to determine our power to detect the odds ratio by simulating our data from study 1 in R based on recommendations from Hsieh [40]. In order to approximate *post hoc* power for this study, we treated every observation as independent (subjects × trials); therefore, *post hoc* power analyses are slightly inflated. However, our actual analyses accounted for both the dependency and independency of our data (see Data analysis section below). Results from the *post hoc* power analysis indicate that we were powered to detect the contact × motivation interaction at 95%. We include confidence intervals for all effects for interested readers.

Additionally, we performed a sensitivity analysis using GPower [41] assuming the achieved alpha significance criterion of the contact × motivation interaction ($p = 0.022$). With a sample size of 168 participants, one tested predictor (the *a priori* predicted interaction), and three total predictors (both main effects and their interaction), we could detect a minimum effect size of $f^2 = 0.059$ with 80% power.

### 2.2.3. Data exclusions

Participants were excluded based on the following criteria: (i) accuracy below chance (less than 25% correct) (1 out of 175 participants); (ii) average reaction time that was three standard deviations faster or slower than the average reaction time within the sample ($M_{reaction\ time} = 5122.34$ s, s.d. = 5608.35 s) (1 out of 175 participants); and (iii) incorrect response to attention check question (6 out of 175 participants, one of whom had already been excluded for below chance accuracy). The final sample size for this study after exclusions was 168 participants ($M_{age} = 32.32$, s.d. = 7.90, 86 female). We also excluded all trials on which participants responded faster or slower than three standard deviations from their average reaction time (excluded 0 out of 6048 total trials, an average of 0 out of 36 trials per participant). Data were reanalysed for consistency with pre-registered exclusion criteria for study 2. See electronic supplementary material, Discussion 2 for original analyses, and see electronic supplementary material, Discussion 1 for pre-registration clarifications regarding exclusion criteria and analyses.

### 2.2.4. Stimuli

This study used the same stimulus set developed by Baron-Cohen *et al.* [10]. This stimulus set consists of 36 greyscale White faces (18 female) cropped using a rectangular area of 466 × 185 pixels to show only the eye region. A set of four emotional words is paired with each face. One of the four words is the target word that best describes the emotional state (e.g. contemplative, fantasizing, etc.) displayed, whereas the other three words are distractors. For a complete list of target words used, see Baron-Cohen *et al.* [10].

## 2.2.5. Experimental protocol

All research was conducted in accordance with the University of Chicago Institutional Review Board.

Stimulus presentation and data collection were completed using Inquisit 4.0 Web (v. 4.0.9; Millisecond Software, Seattle, Washington). Participants were randomly assigned to either the motivation or the control condition. Both groups of participants were instructed to choose the word that best described what the target was thinking or feeling. Participants were also instructed to read all four words before making their choices and encouraged to perform the task as quickly as possible. Following the general instructions, participants in the motivation condition were given the following instructions (adapted from [39]):

> Previous research has shown that as long as people try hard enough, they can significantly improve their performance in perceiving faces. For example, people can be more accurate when recognizing others' emotions if they want to. Now that you know this, we would like you to pay close attention to the faces, and try your best when selecting the word that best describes each person.

In the control condition, participants were not given this additional motivating information. Both groups of participants were then directed to a practice trial where feedback was given. Following this practice trial, no feedback was provided in the experimental task. The face used in the practice trial did not appear elsewhere in the study.

For each of the 36 RME trials, a target word and three distractor words were presented at the four corners of the screen around the centred face image. Word locations were determined based on the original RME test [10]. Participants used the computer mouse to indicate their choices by clicking on the words, and the task was self-paced. Unlike in [10], where every participant received the same order of face presentation, we randomized the face presentation. Following the RME test, participants completed an interracial contact questionnaire [29] and a brief demographic questionnaire.

## 2.2.6. Interracial contact questionnaire

Participants completed an online questionnaire that assessed the composition of their childhood and current social networks across racial groups (i.e. Asian, Black, Hispanic, White and other; [29]). This questionnaire asked participants to report their personal familiarity with outgroup members across several social categories varying in closeness (e.g. friendships, peers, neighbours, etc.) during different stages of their life (0–6 years old, 6–12 years old, 12–18 years old and currently). See the study 2 pre-registration (https://osf.io/v8wgg/, 'Other') for the contact questionnaire in its entirety.

Each participant's average childhood and current contact with Black and White people was calculated respectively, and a difference score between contact with Black versus White people was computed as their average contact with White people subtracted from their average contact with Black people. Thus, each participant had separate childhood and current contact scores that ranged from −100 (0% contact with Black people) to +100 (100% contact with Black people). We used a difference score because we were interested in peoples' contact with Black people relative to their contact with White people among perceivers who typically have tremendous exposure to White faces since childhood. As childhood and current contact were highly correlated ($r_{166} = 0.48$, $p < 0.001$), we computed a composite contact score to index participants' lifetime interracial contact. Specifically, we calculated a measure of lifetime contact by averaging each participant's childhood and current contact difference scores, which we used for all analyses. For analyses exploring childhood and current contact separately and potential interactions between childhood and current contact, see electronic supplementary material, Discussion 3. To maintain consistency with previous research on interracial contact [29], we report analyses that used the Black–White contact difference score; however, for analyses using childhood contact with all non-White races, see electronic supplementary material, Discussion 4.

## 2.2.7. Data analysis

We used mixed-effects logistic regression to analyse these data with the lme4 package [42] in the R programming language [43]. All statistical tests were two-tailed and were based on Satter weights. The dependent variable was trial accuracy (0 = incorrect and 1 = correct). The between-subjects factors were lifetime contact, which was converted to a $z$-score, and the motivation manipulation, which was contrast coded such that −0.5 denoted the control condition and 0.5 denoted the motivation condition. We allowed for between-subjects variance in intercepts to account for variations in response accuracy.

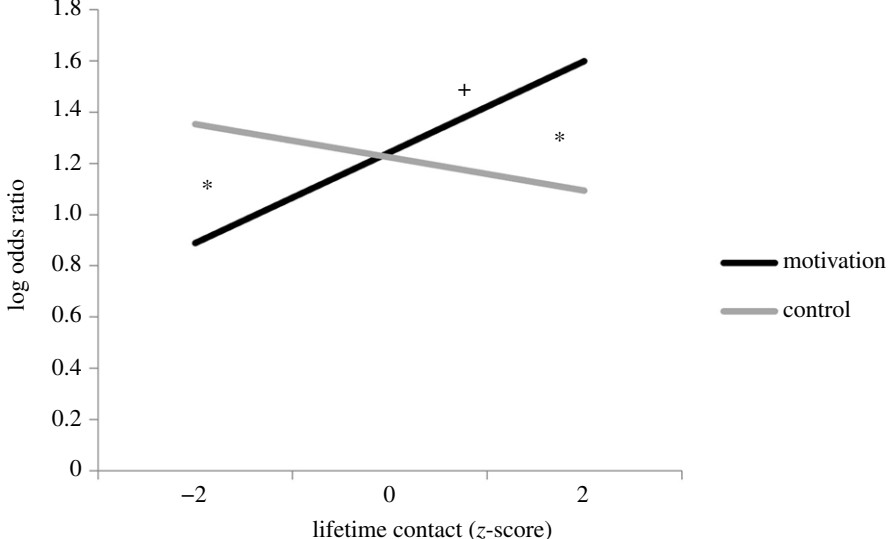

**Figure 1.** Motivation modulated the relationship between lifetime interracial contact and the RME test response accuracy (0 = incorrect trial, 1 = correct trial). Significant differences ($p < 0.05$) between the motivation and control conditions at a given level of contact are indicated with an asterisk. Significant slopes ($p < 0.05$) within each condition are indicated with a cross.

### 2.2.8. Data availability

Data are available on request from the corresponding author.

## 2.3. Results

### 2.3.1. Interracial contact

Overall, participants reported significantly more contact with White than Black individuals throughout their lifetime ($M = -65.44$, s.d. = 21.16), which was confirmed by a one-sample $t$-test comparing mean lifetime contact to 0 (equal contact with Black and White people) ($t_{167} = -40.095$, 95% CI = [−68.699, −62.224], $p < 0.001$). We also ran an independent samples $t$-test to compare lifetime contact scores between the motivation ($M = -65.18$, s.d. = 21.03) and control ($M = -65.70$, s.d. = 21.29) conditions; there were no significant differences in lifetime contact between conditions ($t_{164.27} = -0.149$, 95% CI = [−6.964, 5.984], $p = 0.881$).

### 2.3.2. Regression model

Supporting our predictions, we observed a significant two-way interaction between motivation condition and lifetime interracial contact ($B = 0.242$, s.e. = 0.106, 95% CI = [0.035, 0.449], $z = 2.288$, $p = 0.022$) (figure 1). See table 1 for full regression results including non-significant effects (main effects of motivation and lifetime interracial contact), and see electronic supplementary material, Discussion 5 for data distribution.

We ran follow-up analyses to decompose the interaction between motivation condition and lifetime interracial contact. Because the distribution of interracial contact in the United States tends to be highly negatively skewed and because we were interested in exploring extremes in contact, we chose ±2 s.d. as the *a priori* points for interpreting simple slopes. Elaborating on this analysis decision, we note that most White people tend to have limited contact with racial outgroup individuals but extensive contact with members of their racial ingroup. To focus our analyses on perceivers who have truly meaningfully high amounts of experience with Black people, we specifically wanted to include the +2 s.d. point. To further explore the effects of low and high interracial contact, we re-centred lifetime contact at −2 s.d. below the mean and +2 s.d. above the mean.[1] Low-contact participants in the control condition were significantly more accurate at the RME test than low-contact participants in the motivation condition

---

[1]We note that testing simple slopes using ±1 s.d. is traditional per guidelines proposed by Cohen *et al.* [44]. However, these guidelines were suggested for use in the context of regression models in which no *a priori* meaningful values exist [45]. Based on previous piloting in our laboratory, we selected ±2 s.d. as meaningful points for comparison for both studies; additionally, this decision was explicitly pre-registered in study 2 (https://osf.io/v8wgg/).

**Table 1.** Study 1 regression results predicting whether a participant was correct on a given trial from motivation condition, lifetime interracial contact and their interaction.

| predictors | B | s.e. | 95% CI | z-value | p-value |
|---|---|---|---|---|---|
| intercept | 1.233 | 0.053 | [1.129, 1.338] | 23.093 | <0.001* |
| motivation | 0.020 | 0.106 | [−0.187, 0.228] | 0.192 | 0.848 |
| lifetime interracial contact | 0.056 | 0.053 | [−0.047, 0.160] | 1.063 | 0.288 |
| motivation × lifetime interracial contact | 0.242 | 0.106 | [0.035, 0.449] | 2.288 | 0.022* |

($B = -0.463$, s.e. $= 0.235$, 95% CI $= [-0.924, -0.002]$, $z = -1.967$, $p = 0.049$). This pattern was reversed for high-contact participants, who were significantly more accurate at the RME test in the motivation condition than in the control condition ($B = 0.504$, s.e. $= 0.237$, 95% CI $= [0.039, 0.968]$, $z = 2.124$, $p = 0.034$). The difference in performance between the motivation and control conditions was not significant for participants with average lifetime interracial contact ($B = 0.020$, s.e. $= 0.106$, 95% CI $= [-0.187, 0.228]$, $z = 0.192$, $p = 0.848$).

We also ran models with the motivation and control conditions dummy coded to allow us to predict the effect of lifetime interracial contact on RME test accuracy within each condition. Lifetime contact significantly predicted response accuracy on the RME test in the motivation condition ($B = 0.177$, s.e. $= 0.074$, 95% CI $= [0.033, 0.321]$, $z = 2.406$, $p = 0.016$), but not in the control condition ($B = -0.065$, s.e. $= 0.076$, 95% CI $= [-0.213, 0.084]$, $z = -0.853$, $p = 0.394$). Within the motivation condition, response accuracy on the RME test increased as lifetime interracial contact increased.

## 2.4. Discussion

In our initial investigation, we supported the hypothesis that interracial contact and motivation impact mentalizing. As predicted, consistent with previous work suggesting that greater contact increases face processing efficiency and decreases social cognitive engagement in response to non-salient others [3], high-contact perceivers who were motivated to more deeply process the faces were more accurate at mentalizing than were unmotivated high-contact perceivers. Interestingly, motivation had the opposite effect on low-contact perceivers: motivated low-contact perceivers were significantly less accurate at inferring mental states than unmotivated low-contact perceivers. Although we did not have any *a priori* predictions about the effect of motivation on low-contact perceivers' mentalizing accuracy, it is possible that these perceivers' relatively diminished face processing expertise led to increased explicit monitoring in the motivation condition, which in turn impaired their performance [46,47]. In other words, the motivation manipulation may have led these less expert face processors to 'choke' by leading them to expend cognitive resources to self-monitor instead of focusing fully on the mentalizing task at hand. This is an intriguing possibility for future research to test.

Although study 1 provided initial support for our predictions, previous studies have suggested that contact may preferentially influence the processing of other-race faces [39]. However, recent brain imaging findings indicate an alternative possibility: differences in intergroup contact may influence face processing in a general manner that is not specific to the perceptual target's race [3]. To differentiate between these competing possibilities, in study 2, we investigated mentalizing for both same-race (White) and cross-race (Black) targets among White American participants varying in interracial contact.

# 3. Study 2

## 3.1. Introduction

In study 2, we aimed to replicate and extend findings from study 1 with a highly powered, pre-registered investigation (https://osf.io/v8wgg/). For this study, we created the first Black RME test that was equated with the original Baron-Cohen RME test [10,11]. We investigated the effect of individual differences in lifetime interracial contact and motivation to attend to target faces on White perceivers' ability to infer mental states from same-race (White) and cross-race (Black) faces, predicting that motivation would modulate the relationship between contact and mentalizing ability similarly irrespective of target race.

### 3.1.1. Confirmatory hypotheses

Based on the findings from Cloutier *et al.* [3] demonstrating that changes in recruitment of face processing and social cognitive brain networks associated with contact do *not* vary as a function of target race, we predicted the same pattern of results found in study 1 irrespective of target race (https://osf.io/v8wgg/). In other words, motivated high-contact perceivers would be more accurate than unmotivated high-contact perceivers at inferring both Black and White targets' mental states. We remained agnostic about the effect of motivation on low-contact perceivers' mentalizing accuracy.

### 3.1.2. Exploratory analyses

All exploratory analyses are reported in electronic supplementary material, Discussion 6.

## 3.2. Method

### 3.2.1. Participants

We recruited 788 participants ($M_{age}$ = 32.23, s.d. = 7.70, 458 female) through MTurk, who were paid $2.00 for their participation. Only US-based participants who identified as White European descent (non-Hispanic, non-mixed race), were between the ages of 18 and 50 years old and were born in the United States were eligible to complete the study per our *a priori* inclusion criteria. Eligibility was determined by a brief demographic pre-screen.

### 3.2.2. Power analysis

We include an *a priori* power analysis in the pre-registration of this study (https://osf.io/v8wgg/). However, because of the complicated nature of the design and analyses (two categorical within subjects variables and one between subjects continuous variable, to be analysed using glmer in R) and the fact that study 2 was a conceptual, rather than a direct, replication, *a priori* power analyses were difficult to approximate. Therefore, we used G*Power with an *F*-test statistic to calculate our pre-registered sample size prior to data collection. However, readers should note that because the distribution used was an *F* distribution, sample size estimates required to achieve 80% power could be slightly inflated (i.e. we collected more subjects than was actually required).

We also complement the pre-registered power analysis with a *post hoc* power analysis to determine our power to detect the odds ratio by simulating our data from study 2 in R based on recommendations from Hsieh [40]. In order to approximate *post hoc* power for this study, we treated every observation as independent (subjects × trials); therefore, *post hoc* power analyses may be slightly inflated. However, our actual analyses account for both the dependency and independency of our data (see data analysis section below). Results indicate that we were powered to detect the contact × motivation interaction above 95%. We also include confidence intervals for all effects for interested readers.

Additionally, we performed a sensitivity analysis using GPower [41] assuming the achieved alpha significance criterion of the contact × motivation interaction ($p = 0.015$). With a sample size of 788 participants, one tested predictor (the *a priori* predicted interaction), and seven total predictors (three main effects and all possible interactions), we could detect a minimum effect size of $f^2 = 0.014$ with 80% power.

### 3.2.3. Data exclusions

Based on our pre-registered analysis plan, participants were excluded based on the same criteria as in study 2: (i) accuracy below chance (less than 25% correct) (2 out of 788 participants); (ii) average reaction time that was three standard deviations faster or slower than the average reaction time within the sample ($M_{reaction\ time}$ = 5303.11 s, s.d. = 8548.95 s) (0 out of 788 participants); and/or (iii) incorrect response to at least one out of seven attention check questions, six of which were administered during the RME test and one of which was administered during the contact questionnaire (19 out of 788 participants, two of whom had already been excluded for below chance accuracy). The final sample size after these exclusions was 769 participants. However, in order to reach our pre-registered sample size of 788 participants, we collected data from 19 additional participants to replace the excluded participants resulting in a final sample size of 788 ($M_{age}$ = 32.23, s.d. = 7.70, 458 female). We also

excluded all trials on which participants responded faster or slower than three standard deviations from their individual average reaction time (excluded 853 out of 56 736 total trials (1.50%), an average of 1.08 out of 72 trials per participant). See electronic supplementary material, Discussion 1 for pre-registration clarifications regarding exclusion criteria and analyses.

### 3.2.4. Stimuli

In addition to the 36 White target faces and word choices from the original RME test [10], we developed and equated a set of 36 Black target faces and word choices [11]. This Black stimulus set used the same target emotions and word choices as the original RME task [10,11]. Male and female Black faces portraying different emotional states were collected from movies, music videos, and actors and actresses' headshots online. Faces were converted to greyscale in Adobe Photoshop and cropped using a rectangular area of $466 \times 185$ pixels that delineated the eyes, eyebrow and part of the nose. For a detailed description of the Black RME pilot testing, see electronic supplementary material, Discussion 7 and [11].

### 3.2.5. Experimental protocol

All research was conducted in accordance with the University of Chicago Institutional Review Board.

MTurk participants completed a pre-screening demographics questionnaire and an abbreviated childhood contact screening form. Only participants who met the study's demographic inclusion criteria (i.e. identified as White, between the ages of 18 and 50 years old, born in the United States) continued to the RME test and the follow-up questionnaires. Additionally, to ensure that we had an adequate distribution of contact scores, we screened 200 participants (25.38%) who were required to report a minimum of 15% childhood contact with Black people. The remaining 588 participants could report any amount of childhood contact.

As in study 1, participants completed the RME test through Inquisit 4.0. Participants were randomly assigned to either the motivation or the control condition. The same instructions used to manipulate motivation in study 1 were used in the present study. However, while there were only 36 trials of White faces in study 1, study 2 included 72 trials of pseudo-randomized Black and White faces. Randomization was constrained such that participants did not view more than three Black or White faces in a row. As in study 1, response feedback was only provided for the first practice trial. We recorded reaction time data through Inquisit 4.0, which we used for exploratory analyses only (see electronic supplementary material, Discussion 6). Following the RME test, participants completed the unabbreviated interracial contact questionnaire [29] and the modern racism scale (MRS; [48]) (see electronic supplementary material, Discussion 6 for exploratory analyses with MRS).

### 3.2.6. Interracial contact questionnaire

The same contact questionnaire described in study 1 was administered in study 2. See the pre-registration (https://osf.io/v8wgg/, 'Other') to view the contact questionnaire in its entirety. As in study 1, each participant's average childhood and current contact with Black people and White people was calculated respectively, and a Black–White difference score was then computed for childhood and current contact, respectively ($M_{\text{Black contact}} - M_{\text{White contact}}$). Similar to study 1, childhood and current contact were highly correlated ($r_{786} = 0.623$, $p < 0.001$), so we calculated a measure of lifetime contact by averaging each participant's childhood and current contact difference scores, which we used for all analyses. For analyses exploring childhood and current contact separately and potential interactions between childhood and current contact, see electronic supplementary material, Discussion 3. As in study 2, we focused our analyses on contact with Black versus White individuals; however, for analyses that include contact with all races see electronic supplementary material, Discussion 4.

### 3.2.7. Data analysis

We used mixed-effects logistic regression to analyse these data with the lme4 package [42] in the R programming language [43]. All statistical tests were two-tailed and were based on Satter weights. The dependent variable was trial accuracy (0 = incorrect and 1 = correct). The within-subjects factor was target race ($-0.5$ = Black targets and $0.5$ = White targets). The between-subjects factors were lifetime contact, which was converted to a $z$-score, and the motivation manipulation, which was contrast coded ($-0.5$ = control and $0.5$ = motivation). We allowed for between-subjects variance in

intercepts and slopes as a function of target race (i.e. random effects) to account for participant variations in response accuracy as a function of target race.

### 3.2.8. Data availability

Data are available on request from the corresponding author.

## 3.3. Results

### 3.3.1. Individual difference measures

As in study 1, participants' social networks across their lifespan included more White people than Black people ($M_{\text{lifetime contact}} = -54.92$, s.d. = 26.00), as confirmed through a one-sample $t$-test comparing mean lifetime contact against 0 ($t_{787} = -59.301$, 95% CI = [$-56.740$, $-53.104$], $p < 0.001$). There was a small but significant difference in lifetime interracial contact between the motivation ($M_{\text{lifetime contact}} = -57.20$, s.d. = 24.73) and control ($M_{\text{lifetime contact}} = -52.65$, s.d. = 27.05) conditions ($t_{779.72} = 2.465$, 95% CI = [$-52.646$, $-57.197$], $p = 0.014$). Because of this difference, we partialled out the variance in lifetime interracial contact associated with the motivation and control conditions and re-ran all analyses. When accounting for the difference in contact between conditions, all significant results held and no non-significant results became significant, suggesting that the difference in lifetime contact between the motivation and control condition could not explain our pattern of results.

### 3.3.2. Confirmatory analyses

Because the revised RME task in study 2 had twice the number of trials as the original RME task used in study 1 due to the inclusion of both same- and cross-race faces, participants could have shown diminished attention and impaired performance towards the end of the task. However, performance across conditions did not differ between the first half and the second half of the RME task (see electronic supplementary material, Discussion 1). Thus, analyses were performed on all 72 trials.

Results revealed significant main effects of target race ($B = -0.143$, s.e. = 0.020, 95% CI = [$-0.183$, $-0.103$], $z = -7.07$, $p < 0.001$), and lifetime contact ($B = -0.080$, s.e. = 0.019, 95% CI = [$-0.117$, $-0.043$], $z = -4.20$, $p < 0.001$); however, the main effect of lifetime contact was qualified by the predicted two-way interaction of motivation and lifetime contact ($B = 0.092$, s.e. = 0.038, 95% CI = [0.018, 0.167], $z = 2.42$, $p = 0.015$). See table 2 for full regression results including all non-significant predictors, and see electronic supplementary material, Discussion 5 for data distribution.

### 3.3.3. Main effect of target race

In the logistic regression model, there was a significant main effect of target race on RME test accuracy ($B = -0.143$, s.e. = 0.020, 95% CI = [$-0.183$, $-0.103$], $z = -7.07$, $p < 0.001$). We did not find an own-race advantage for inferring mental states; instead, we found that White participants were significantly more accurate at inferring mental states from Black target faces ($M_{\text{accuracy}} = 0.752$, s.d. = 0.432) than from White target faces ($M_{\text{accuracy}} = 0.725$, s.d. = 0.447). However, target race did not significantly interact with any other predictors.

### 3.3.4. Interaction between motivation condition and lifetime interracial contact

There was also a significant main effect of lifetime interracial contact on RME test accuracy, suggesting that low-contact participants were more accurate at the RME test than high-contact participants. However, the main effect of lifetime interracial contact was qualified by the predicted significant two-way interaction between motivation and lifetime interracial contact ($B = 0.092$, s.e. = 0.038, 95% CI = [0.018, 0.167], $z = 2.42$, $p = 0.015$; figure 2).

We ran planned follow-up analyses to interpret the significant interaction of motivation condition and lifetime interracial contact. As in study 1, we re-centred lifetime contact at $-2$ s.d. below the mean and $+2$ s.d. above the mean to interpret the effect of the motivation manipulation among low- and high-contact participants, respectively. As in both previous studies, the decision to use ±2 s.d. was made *a priori* and was explicitly pre-registered for the present study (https://osf.io/v8wgg/, 'Transformations'; see footnote 1). Our analyses for low- (max | grad | = 0.00172, tol = 0.001) and high-contact (max | grad | = 0.00114, tol = 0.001) participants did not converge due to small cell sizes within our random factor of

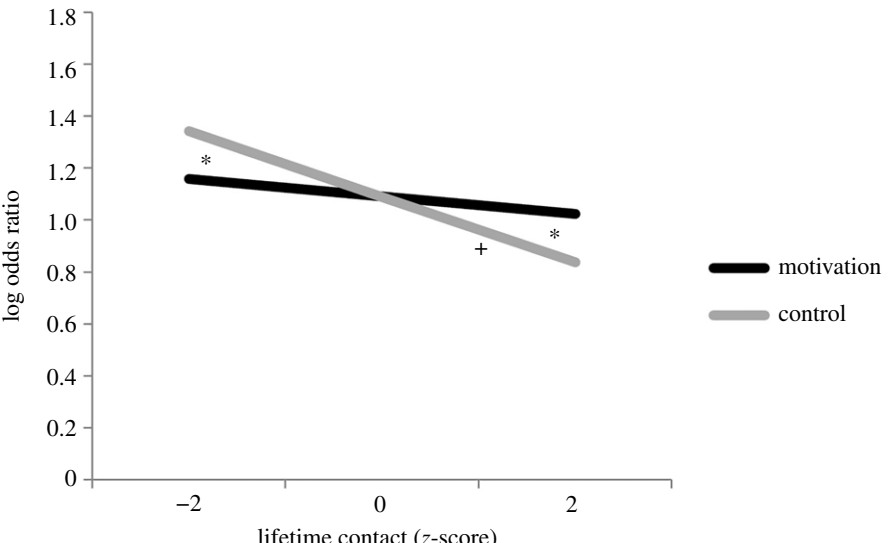

**Figure 2.** Motivation modulated the relationship between lifetime interracial contact and the RME test response accuracy (0 = incorrect trial, 1 = correct trial) irrespective of target race. Significant differences ($p < 0.05$) between the motivation and control conditions at a given level of contact are indicated with an asterisk. Significant slopes ($p < 0.05$) within each condition are indicated with a cross.

**Table 2.** Study 2 regression results predicting mentalizing accuracy on a given trial from target race, motivation condition, lifetime interracial contact and all possible interactions.

| predictors | B | s.e. | 95% CI | z-value | p-value |
| --- | --- | --- | --- | --- | --- |
| intercept | 1.090 | 0.019 | [1.052, 1.127] | 57.16 | <0.001* |
| target race | −0.143 | 0.020 | [−0.183, −0.103] | −7.07 | <0.001* |
| motivation | 0.001 | 0.038 | [−0.074, 0.075] | 0.02 | 0.984 |
| lifetime interracial contact | −0.080 | 0.019 | [−0.117, −0.043] | −4.20 | <0.001* |
| target race × motivation | 0.042 | 0.039 | [−0.035, 0.120] | 1.07 | 0.286 |
| target race × lifetime interracial contact | 0.005 | 0.020 | [−0.033, 0.043] | 0.26 | 0.795 |
| motivation × lifetime interracial contact | 0.092 | 0.038 | [0.018, 0.167] | 2.42 | 0.015* |
| target race × motivation × lifetime interracial contact | 0.031 | 0.039 | [−0.046, 0.107] | 0.79 | 0.428 |

target race at the low- and high-contact extremes; however, removing the target race random factor from the models allowed them to converge and did not change the results. We therefore report results from these non-convergent models. Replicating our findings from study 1, low-contact participants in the control condition were significantly more accurate at the RME test than low-contact participants in the motivation condition ($B = -0.183$, s.e. = 0.085, 95% CI = [−0.350, −0.017], $z = -2.16$, $p = 0.031$). This pattern was again reversed for high-contact participants, who were significantly more accurate at the RME test in the motivation condition than in the control condition ($B = 0.185$, s.e. = 0.085, 95% CI = [0.018, 0.351], $z = 2.18$, $p = 0.030$). The difference in performance between the motivation and control conditions was not significant for participants with average differences in lifetime interracial contact ($B = 0.001$, s.e. = 0.038, 95% CI = [−0.074, 0.075], $z = 0.02$, $p = 0.984$).

We also ran models with the motivation and control conditions dummy coded to allow us to predict the effect of lifetime interracial contact on RME test accuracy within these conditions. Unlike in study 1, in this well-powered pre-registered analysis, lifetime contact did not significantly predict response accuracy on the RME test in the motivation condition ($B = -0.034$, s.e. = 0.028, 95% CI = [−0.089, 0.021], $z = -1.20$, $p = 0.229$). However, in the control condition, lifetime contact significantly predicted response accuracy on the RME test ($B = -0.126$, s.e. = 0.026, 95% CI = [−0.176, −0.076], $z = -4.92$, $p < 0.001$). Within the control condition, response accuracy on the RME test decreased as lifetime interracial contact increased.

## 3.4. Discussion

We partially replicated the effects of study 1 in a pre-registered highly powered investigation. Specifically, we found that motivation modulated the relationship between interracial contact and mentalizing irrespective of target race. As in study 1, and as predicted, high-contact perceivers were more accurate at mentalizing when motivated to deeply process others than when they were not, and the opposite was again true of low-contact perceivers. Although study 2 replicated the expected pattern of simple effects from study 1, unexpectedly, the slopes within each condition differed in this pair of studies. In study 1, mentalizing accuracy *increased* as contact increased in the *motivation* condition, whereas there was no significant relationship between contact and mentalizing accuracy in the control condition. However, in study 2, mentalizing accuracy *decreased* as contact increased in the *control* condition whereas there was no significant relationship between contact and mentalizing accuracy in the motivation condition. Despite this difference, we note that the direction and significance of the slopes in both cases (especially when paired with the remarkably consistent pattern of simple effects) is consistent with the argument that high-contact perceivers may be particularly efficient face processors who require specific motivation to process faces more deeply and thus mentalize accurately. Importantly, we did not find evidence for any interactions between interracial contact and target race, further corroborating the results of study 1 and previous researchers' findings that contact seems to be associated with *general* changes in social cognitive engagement regardless of target race [3].

## 4. General discussion

To our knowledge, this is the first investigation demonstrating that individual differences in interracial contact are associated with changes in White perceivers' mental states inference accuracy from White and Black facial expressions displaying secondary emotions, a critical component of everyday social interactions. The present work extends our knowledge of the social cognitive abilities (i.e. categorization [22,49], person perception and evaluation [50–54], and memory ([24,25,55–58]; Sporer [59,60])) that may be influenced by intergroup contact to include mentalizing. Notably, using the first RME task that included both Black and White perceptual targets, we found that contact shaped White participants' mentalizing similarly for same- and cross-race targets, supporting the possibility that contact as a form of general (i.e. not race-specific) face expertise increases processing efficiency. High-contact perceivers were more accurate at inferring mental states only after being motivated to attend to the task. Somewhat unexpectedly, motivation consistently had the opposite effect on low-contact perceivers: low-contact perceivers were less accurate at inferring mental states when they were motivated than when they were not, perhaps reflecting their diminished face processing expertise leading to excessive explicit monitoring at the expense of accurate mentalizing under the motivation condition (i.e. 'choking'; [46,47]).

Why might greater contact result in decreased recruitment of regions involved in mentalizing? Social cognitive models of intergroup face processing suggest that experience with diverse face exemplars may render faces less distinctive by shifting the perceiver's representation of overall 'average' faces away from a prototypical same-race face and by increasing the extent to which variation from this average is perceived to belong within a normal range of faces (e.g. [61–63]). In line with these models, high-contact perceivers' decreased recruitment of brain regions involved in mentalizing during impression formation may be the result of increased face processing efficiency (i.e. this outgroup face is 'normal' and does not require deeper processing). Indeed, increased processing efficiency can lead to decreased spontaneous social cognitive engagement particularly when there is a reduced need to effortfully process faces that deviate from the normal range typically encountered. Phrased another way, diverse face processors (i.e. those with high levels of previous interracial contact) may be less likely to spontaneously engage in mentalizing because their face space (e.g. [63]) is more inclusive and variations from their perceived average face are less distinctive and less salient to them [3]. Although perhaps counterintuitive, this explanation fits well with our finding that high-contact perceivers are less accurate at mentalizing about others when they are not specifically motivated to do so. Increased social cognitive efficiency does not necessarily result in worse mentalizing in and of itself; however, these more efficient processors may get away with expending less cognitive effort when inferring mental states in relatively low stakes situations such as the RME task.

At first glance, the possible association between increased contact and decreased accuracy when inferring mental states appears at odds with findings from studies on the cross-race recognition

deficit. Specifically, we argued that outgroup faces are less socially salient to high-contact individuals and therefore require less processing. However, findings from cross-race recognition deficit studies suggest that ingroup faces—the faces with whom most individuals have the vast majority of their experience and are therefore the most 'typical' faces one can encounter—are better remembered than outgroup faces (for review, see [54]). In response to this apparent contradiction, we note that there are important differences between the task components involved in face recognition and those involved in inferring a mental state based on someone's facial expressions. Notably, the RME task does not require the encoding, storage and retrieval of information about a face, but instead involves a relatively effortful explicit online mental state judgement based on facial features. These differences may underlie our diverging results, although future work should directly test this explanation. Studies manipulating other-race experience (rather than measuring contact, as we have done in the present work) may provide a particularly promising avenue for exploring this question; we discuss such studies in the Limitations and future directions section.

Although it was not the primary objective of the study, the overall difference in target race accuracy in study 2 was not predicted (note target race did not modulate any other effects). The direction of this effect, wherein White perceivers were more accurate at inferring mental states from Black than White eyes, may seem particularly surprising given past work on cross-race emotion recognition (e.g. [64–67]) and that the Black RME targets were specifically equated on difficulty with the White RME targets [11]. Although the current data cannot fully explain this finding, one possible explanation for this effect is that our White participants were more accurate at inferring mental states from Black targets because these targets' emotional displays were perceived as more relevant to these participants. This possibility is consistent with previous work showing that the cross-race recognition deficit is eliminated when White people attempt to recall angry Black faces [68]. Future research is needed to explore how racial and cultural group membership influences both primary and secondary emotion recognition.

With regard to the RME in particular, it is also noteworthy that this is the first study to examine the relationship between interracial contact and within-culture cross-race mentalizing (rather than cross-cultural mentalizing; see [7] and [9]). It is also the first study to use an interleaved rather than blocked presentation of Black and White mentalizing targets. The Black RME stimuli were piloted in a between-subjects design (i.e. Black and White participants either completed the RME with White targets *or* the RME with Black targets, but not both). When stimuli were not presented in random order, mentalizing accuracy did not differ by target race [11]. In our investigation, this randomized presentation of Black and White eyes may have made target race more salient. Because societal norms generally encourage people to be (or at least appear to be) racially egalitarian [69], it is possible that White participants were particularly motivated to attend to Black targets over White targets. Future research should address this question directly.

## 4.1. Limitations and future directions

Although the current research provides the foundation for future research on contact and mentalizing, the use of a self-report measure of interracial contact prevents us from assessing a causal relationship between increased contact and mentalizing. The correlational nature of the present work does not allow us to rule out the possibility that confounds such as socioeconomic status and/or education may have influenced the relationship between interracial contact and mentalizing accuracy. Although in our previous contact studies we have found that our effects are not explained by population density, future work including additional control variables such as geographical region of residence will be important to confirm the robustness of these findings [3]. Additionally, although meaningful intergroup contact can be difficult to experimentally manipulate in the laboratory, contact interventions and training procedures have been employed to that purpose [53,70–74]. As such, future research should aim to experimentally manipulate intergroup experience to explore its possible causal impact on mentalizing performance.

This research primarily uses the face perception literature to derive our hypotheses as some previous research indicates that face processing effects are mediated by the eye region (e.g. [75,76]). However, it remains unclear whether our results would generalize to situations where individuals view the whole face where they have access to additional featural cues. For example, research suggests that individuals fixate on both the mouth and eyes when processing happy faces while individuals primarily rely on the eyes for processing angry faces (e.g. [77]). Therefore, in order to generalize these findings, future research should directly compare how contact shapes mentalizing when individuals view the eye region only versus when they view the whole face.

It is important to reiterate that both of the present studies used exclusively White participants. Our reasoning for this decision was to minimize between-subjects variability and provide greater statistical power to detect individual difference (i.e. contact) effects on mentalizing for this initial investigation of the phenomenon. Additionally, contact between White and Black individuals in the United States has a unique history and a distinct contemporary social context that may not hold true for other races; it is not clear that other-race individuals (e.g. Hispanic or Asian) inferring mental states from White and Black faces would experience the same social cognitive processes as White people do. The present studies aim to lay the groundwork for future work to investigate these important questions. However, given that the present work found that contact shaped inferences of mental states based on facial expressions regardless of the race of the face, it is critically important that future work test whether this generalizes to perceivers from other racial groups or whether this effect is limited to European American perceivers. Future research should also probe whether these results are obtained when measuring substantial contact with multiple different racial groups across the lifespan rather than the relative Black/White contact measure used in the present work.

Finally, the contact questionnaire used in both studies provides a holistic measure of contact; it does not distinguish between quality and quantity of contact, nor does it lend itself to the assessment of population density or the amount of interaction with people in general. Previous research has demonstrated that population density (i.e. simply encountering more faces) does not account for the observed decrease in social cognitive engagement among high-contact perceivers [3]. Therefore, we would not anticipate changes in our findings as a function of population density. The distinction between quality and quantity of contact is difficult to disentangle because both forms of contact tend to be highly correlated ([78]; Sporer [59]); however, this distinction has been shown to be important in studies of affective bias and intergroup contact and represents an important future research direction [16,70,79–82].

Although future work will be needed to clarify these outstanding questions, the present research represents an important first step toward understanding how interracial contact influences mentalizing from both same- and cross-race faces. These findings are the first to demonstrate that interracial contact, as a form of *general* face processing expertise, may be associated with similar changes in mentalizing about both racial outgroup and ingroup members.

# 5. Conclusion

The present study extends our understanding of intergroup mentalizing, finding that contact and motivation jointly affects perceivers' mentalizing ability for both same- *and* cross-race faces. Whereas much of the vast literature on intergroup contact has focused on its important role in minimizing racial biases and prejudice [2], the current findings illustrate the broader impact that our increasingly racially diverse environments may have on social cognition irrespective of the race of others. These findings provide initial evidence that opportunities for increasing contact may have broader implications for social interactions than previously posited.

# 6. Open practices

In these studies, we report all measures, manipulations and exclusions. Sample size was determined before any data analysis. Study data are available through Mendeley Data. Supplemental data and study materials are available on request from the corresponding author. Study 2 was pre-registered through the Open Science Framework. The pre-registration is available at https://osf.io/v8wgg/.

Ethics. All research was conducted in accordance with the University of Chicago Institutional Review Board (IRB15-0301).
Data accessibility. Data and analysis scripts for studies 1 and 2 are available at https://osf.io/p4a3c/.
Authors' contributions. G.H. and J.T.K. contributed equally to this work. G.H., J.T.K., T.L. and J.C. developed the study concept and contributed to the study design. Programming, testing and data collection were conducted by G.H. and T.L. G.H. performed the data analysis and interpretation with guidance from J.T.K. and J.C. G.H. drafted the manuscript, and J.T.K., T.L. and J.C. provided critical revisions. All authors approved the final version of the manuscript for submission.
Competing interests. We declare we have no competing interests.
Funding. We received no funding for this study.
Acknowledgements. We would like to thank Tanisha Jain for their assistance in the early stages of participant recruitment, stimulus piloting, and data collection.

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
