## [Peer Review File · Royal Society Open Science]

Review History

RSOS-202137.R0 (Original submission)

Review form: Reviewer 1

Is the manuscript scientifically sound in its present form?

Yes

Are the interpretations and conclusions justified by the results?

Yes

Is the language acceptable?

Yes

Do you have any ethical concerns with this paper?

No

Have you any concerns about statistical analyses in this paper?

No

Recommendation?

Major revision is needed (please make suggestions in comments)

Comments to the Author(s)

The present paper explores an interesting topic: that of the effect of lifetime interracial contact and motivation to be accurate on mentalizing as measured by Reading the Mind in the Eyes (RME) tests that included both same- and cross-race target faces. In a first study, White control participants with more lifetime interracial contact were less accurate on the RME that included only White target faces. This effect was reversed in a motivation condition such that lifetime contact increased accuracy among motivated White participants. In a second study, White participants performed the RME with both White and Black target faces. The relationship between lifetime interracial contact and accuracy for control participants was again negative. However, while motivated participants with extensive lifetime contact were more accurate than controls, there was not a positive relationship between lifetime contact and accuracy overall in the motivated condition.

These are interesting findings, and the analyses seem competent. Some work on the writing itself would be necessary for it to have the desired and warranted scientific impact:

1. The introduction sets up Study 1 less well than it should. The reasoning is that interracial contact increases face processes efficiency overall and that this can lead to worse mentalizing overall (i.e., independent of same- or cross-race status of a target face). And the purpose of Study 1 is to demonstrate this general effect of contact on same-race faces and also show that motivation to decode correctly increases mentalizing accuracy for these same-race targets. (Of course, only White participants and target faces are used here, so the generalizability is not great.) However, the goals of Study 1 do not follow naturally enough from the introduction. This is because most of the cited work is on cross-race perception and mentalizing, not on the effect of contact on mentalizing accuracy overall. Since the two effects are not separated clearly enough, there is not a clear and natural progression to the design of Study 1. There is an attempt to do that in the overview paragraph, but this comes too late.
2. The relationship to lifetime contact is of course correlational and this was noted in the discussion of the study's limitations. However, usually when a variable that is not manipulated is used as a significant predictor, a number of other control variables are entered into the model in order to enhance the ability to draw conclusions about that one variable. I'm wondering therefore why other variables such as age were not used as controls for things like historical effects on racial attitudes. Region of the US would have also been a helpful control variable as interracial contact may overall be different in the North and the South and these may be confounded in the contact measure.
3. I think that the lengthy discussion of the cross-race finding (greater accuracy by Whites in decoding expressions of Black faces on the RME test) in discussing the findings of Study 2 is premature. Since the target faces in the White and Black RME tests were those of different people altogether, it could be that the Black targets for any number of different reasons were better at encoding the expressions in the first place. This interpretation is not necessarily undermined by the fact that a mentioned pilot test didn't find the same main effect (or at least I do not know enough about the pilot test to believe that the new finding is the is something worth interpreting as it is here).

Review form: Reviewer 2

Is the manuscript scientifically sound in its present form?

Yes

Are the interpretations and conclusions justified by the results?

Yes

Is the language acceptable?

Yes

Do you have any ethical concerns with this paper?

No

Have you any concerns about statistical analyses in this paper?

No

Recommendation?

Accept with minor revision (please list in comments)

Comments to the Author(s)

The current manuscript reports two studies examining how intergroup contact influences the ability to infer others' mental states.

The paper is well written and the literature proposed to support Authors' viewpoint is adequate. Also, statistical analyses are sound. The results are interesting and support the conclusions.

I have some points that the Authors should consider in revising their work.

Firstly, the Authors introduced their studies by summarizing intergroup contact literature with some references to neuroimaging studies. However, the evidence supporting this argument is rather limited at this point. I think it could be useful to delve a little deeper into them, by considering recent reviews such as Han, 2018 TICs, Bagnis et al. 2019 Neurosci. Biobehav. Rev, Bagnis et al. 2020 Neuroimage.

Also, authors should specify in the introduction that they used two different versions of the RME, i.e. the original one (Baron-Cohen et al., 2001) and the Black RME (Handley et al., 2019).

Finally, I am not sure if I follow the authors' logic of using intergroup face processing models to explain some of their results (p.28). As they state later (p.29), the RME task does not involve the processing of a whole face.

Decision letter (RSOS-202137.R0)

Dear Miss Handley,

The Editors assigned to your paper RSOS-202137 "Impact of interracial contact on inferring mental states from facial expressions" have now received comments from reviewers and would like you to revise the paper in accordance with the reviewer comments and any comments from the Editors. Please note this decision does not guarantee eventual acceptance.

Please submit your revised manuscript and required files (see below) no later than 21 days from today's (ie 08-Feb-2021) date. Note: the ScholarOne system will 'lock' if submission of the revision is attempted 21 or more days after the deadline. If you do not think you will be able to meet this deadline please contact the editorial office immediately.

on behalf of Dr Giorgia Silani (Associate Editor) and Essi Viding (Subject Editor)
openscience@royalsociety.org

Associate Editor Comments to Author (Dr Giorgia Silani):

While the reviewers found your work to be potentially important and conceptually appropriate for RSOS, they have highlighted some weakness and provided constructive suggestions that would need to be addressed before the manuscript would be considered for publication. Thus, I would be glad to reconsider a revised manuscript which takes into account the points raised by the reviewers.

Reviewer comments to Author:

Reviewer: 1

Comments to the Author(s)

The present paper explores an interesting topic: that of the effect of lifetime interracial contact and motivation to be accurate on mentalizing as measured by Reading the Mind in the Eyes (RME) tests that included both same- and cross-race target faces. In a first study, White control participants with more lifetime interracial contact were less accurate on the RME that included only White target faces. This effect was reversed in a motivation condition such that lifetime contact increased accuracy among motivated White participants. In a second study, White participants performed the RME with both White and Black target faces. The relationship between lifetime interracial contact and accuracy for control participants was again negative. However, while motivated participants with extensive lifetime contact were more accurate than controls, there was not a positive relationship between lifetime contact and accuracy overall in the motivated condition.

These are interesting findings, and the analyses seem competent. Some work on the writing itself would be necessary for it to have the desired and warranted scientific impact:

1. The introduction sets up Study 1 less well than it should. The reasoning is that interracial contact increases face processes efficiency overall and that this can lead to worse mentalizing overall (i.e., independent of same- or cross-race status of a target face). And the purpose of Study 1 is to demonstrate this general effect of contact on same-race faces and also show that motivation to decode correctly increases mentalizing accuracy for these same-race targets. (Of course, only White participants and target faces are used here, so the generalizability is not great.) However, the goals of Study 1 do not follow naturally enough from the introduction. This is because most of the cited work is on cross-race perception and mentalizing, not on the effect of contact on mentalizing accuracy overall. Since the two effects are not separated clearly enough, there is not a clear and natural progression to the design of Study 1. There is an attempt to do that in the overview paragraph, but this comes too late.
2. The relationship to lifetime contact is of course correlational and this was noted in the discussion of the study's limitations. However, usually when a variable that is not manipulated is used as a significant predictor, a number of other control variables are entered into the model in order to enhance the ability to draw conclusions about that one variable. I'm wondering therefore why other variables such as age were not used as controls for things like historical effects on racial attitudes. Region of the US would have also been a helpful control variable as interracial contact may overall be different in the North and the South and these may be confounded in the contact measure.
3. I think that the lengthy discussion of the cross-race finding (greater accuracy by Whites in decoding expressions of Black faces on the RME test) in discussing the findings of Study 2 is premature. Since the target faces in the White and Black RME tests were those of different people altogether, it could be that the Black targets for any number of different reasons were better at encoding the expressions in the first place. This interpretation is not necessarily undermined by the fact that a mentioned pilot test didn't find the same main effect (or at least I do not know enough about the pilot test to believe that the new finding is the is something worth interpreting as it is here).

Reviewer: 2

Comments to the Author(s)

The current manuscript reports two studies examining how intergroup contact influences the ability to infer others' mental states.

The paper is well written and the literature proposed to support Authors' viewpoint is adequate.

Also, statistical analyses are sound. The results are interesting and support the conclusions.

I have some points that the Authors should consider in revising their work.

Firstly, the Authors introduced their studies by summarizing intergroup contact literature with some references to neuroimaging studies. However, the evidence supporting this argument is rather limited at this point. I think it could be useful to delve a little deeper into them, by considering recent reviews such as Han, 2018 TICS, Bagnis et al. 2019 Neurosci. Biobehav. Rev, Bagnis et al. 2020 Neuroimage.

Also, authors should specify in the introduction that they used two different versions of the RME, i.e. the original one (Baron-Cohen et al., 2001) and the Black RME (Handley et al., 2019).

Finally, I am not sure if I follow the authors' logic of using intergroup face processing models to explain some of their results (p.28). As they state later (p.29), the RME task does not involve the processing of a whole face.

===PREPARING YOUR MANUSCRIPT===

===PREPARING YOUR REVISION IN SCHOLARONE===

- An individual file of each figure (EPS or print-quality PDF preferred [either format should be produced directly from original creation package], or original software format).
 - An editable file of each table (.doc, .docx, .xls, .xlsx, or .csv).
 - An editable file of all figure and table captions.
- Note: you may upload the figure, table, and caption files in a single Zip folder.
- Any electronic supplementary material (ESM).
 - If you are requesting a discretionary waiver for the article processing charge, the waiver form must be included at this step.
 - If you are providing image files for potential cover images, please upload these at this step, and inform the editorial office you have done so. You must hold the copyright to any image provided.
 - A copy of your point-by-point response to referees and Editors. This will expedite the preparation of your proof.

- Ensure that your data access statement meets the requirements at <https://royalsociety.org/journals/authors/author-guidelines/#data>. You should ensure that you cite the dataset in your reference list. If you have deposited data etc in the Dryad repository, please include both the 'For publication' link and 'For review' link at this stage.
- If you are requesting an article processing charge waiver, you must select the relevant waiver option (if requesting a discretionary waiver, the form should have been uploaded at Step 3 'File upload' above).
- If you have uploaded ESM files, please ensure you follow the guidance at <https://royalsociety.org/journals/authors/author-guidelines/#supplementary-material> to include a suitable title and informative caption. An example of appropriate titling and captioning may be found at https://figshare.com/articles/Table_S2_from_Is_there_a_trade-off_between_peak_performance_and_performance_breadth_across_temperatures_for_aerobic_sc_ope_in_teleost_fishes_/3843624.

Author's Response to Decision Letter for (RSOS-202137.R0)

See Appendix A.

RSOS-202137.R1 (Revision)

Review form: Reviewer 2

Is the manuscript scientifically sound in its present form?

Yes

Are the interpretations and conclusions justified by the results?

Yes

Is the language acceptable?

Yes

Do you have any ethical concerns with this paper?

No

Have you any concerns about statistical analyses in this paper?

Yes

Recommendation?

Accept as is

Comments to the Author(s)

The authors have replied to my previous comments and amended the ms accordingly. I do not have additional comments to offer to what I think is a ms in good shape for publication

Decision letter (RSOS-202137.R1)

Dear Miss Handley,

It is a pleasure to accept your manuscript entitled "Impact of interracial contact on inferring mental states from facial expressions" in its current form for publication in Royal Society Open Science. The comments of the reviewer(s) who reviewed your manuscript are included at the foot of this letter.

Please ensure that you send to the editorial office individual files for each figure and table included in your manuscript. You can send these in a zip folder if more convenient. Failure to provide these files may delay the processing of your proof.

on behalf of Dr Giorgia Silani (Associate Editor) and Essi Viding (Subject Editor)
openscience@royalsociety.org

Reviewer comments to Author:
Reviewer: 2

Comments to the Author(s)

The authors have replied to my previous comments and amended the ms accordingly. I do not have additional comments to offer to what I think is a ms in good shape for publication

Appendix A

Associate Editor Comments to Author (Dr Giorgia Silani):

While the reviewers found your work to be potentially important and conceptually appropriate for RSOS, they have highlighted some weakness and provided constructive suggestions that would need to be addressed before the manuscript would be considered for publication. Thus, I would be glad to reconsider a revised manuscript which takes into account the points raised by the reviewers.

We appreciate the thoughtful and thorough feedback from yourself and the reviewers. We have prepared this revised manuscript according to this feedback, and we hope you will find this revised manuscript to be stronger, clearer, and more specific as a result. We respond to all comments line-by-line in the following letter.

Reviewer comments to Author:

Reviewer: 1

Comments to the Author(s)

The present paper explores an interesting topic: that of the effect of lifetime interracial contact and motivation to be accurate on mentalizing as measured by Reading the Mind in the Eyes (RME) tests that included both same- and cross-race target faces. In a first study, White control participants with more lifetime interracial contact were less accurate on the RME that included only White target faces. This effect was reversed in a motivation condition such that lifetime contact increased accuracy among motivated White participants. In a second study, White participants performed the RME with both White and Black target faces. The relationship between lifetime interracial contact and accuracy for control participants was again negative. However, while motivated participants with extensive lifetime contact were more accurate than controls, there was not a positive relationship between lifetime contact and accuracy overall in the motivated condition.

These are interesting findings, and the analyses seem competent. Some work on the writing itself would be necessary for it to have the desired and warranted scientific impact:

1. The introduction sets up Study 1 less well than it should. The reasoning is that interracial contact increases face processes efficiency overall and that this can lead to worse mentalizing overall (i.e., independent of same- or cross-race status of a target face). And the purpose of Study 1 is to demonstrate this general effect of contact on same-race faces and also show that motivation to decode correctly increases mentalizing accuracy for these same-race targets. (Of course, only White participants and target faces are used here, so the generalizability is not great.) However, the goals of Study 1 do not follow naturally enough from the introduction. This is because most of the cited work is on cross-race perception and mentalizing, not on the effect of contact on mentalizing accuracy overall. Since the two effects are not separated clearly enough, there is not a clear and natural progression to the design of Study 1. There is an attempt to do that in the overview paragraph, but this comes too late.

We appreciate this feedback and agree that the rationale for Study 1 should be set up earlier on and more clearly in the introduction. To that end, we have added the following paragraph that we believe better sets up our approach in general and Study 1 in particular:

“Given that little work to date has tested how interracial contact specifically impacts mental state inferences about others, we present two studies examining how interracial contact shapes White perceivers’ ability to infer mental states from only same-race targets (i.e., White individuals; Study 1) and from same-race and cross-race targets (i.e., White and Black individuals; Study 2). Although interracial contact has

typically been studied in the context of how it impacts the perception and evaluation of cross-race targets – work which we highlight later – previous neuroimaging research showing that increased contact shapes the recruitment of brain networks involved in face processing and social cognition *regardless of target race* (Cloutier et al., 2017) led us to expect that contact may also impact mental state inferences from same-race targets. Thus, Study 1 tested only same-race targets. In Study 2 we extended this work to include both same-race and cross-race targets based on the extensive body of work suggesting contact shapes how cross-race targets are perceived.” (pages 3-4)

2. The relationship to lifetime contact is of course correlational and this was noted in the discussion of the study's limitations. However, usually when a variable that is not manipulated is used as a significant predictor, a number of other control variables are entered into the model in order to enhance the ability to draw conclusions about that one variable. I'm wondering therefore why other variables such as age were not used as controls for things like historical effects on racial attitudes. Region of the US would have also been a helpful control variable as interracial contact may overall be different in the North and the South and these may be confounded in the contact measure.

This is a valuable point. Although unfortunately we do not have data to include these control variables in our models for these studies, we agree that testing these effects with additional control variables is an important future direction. In additional work examining the effect of contact by our group (Cloutier et al., 2017; Handley et al., under review), we have included population density as a control variable (i.e., how do we know it's experience with other-race faces specifically and not just faces in general that drives some of these effects?) and have found that our contact effects are consistently not explained by population density. Therefore, we are at least preliminarily optimistic the contact effects we report should remain robust in the presence of these additional controls. We now mention this important future direction to the discussion noting this previous work and the need for future work including additional control variables: "Although in our previous contact studies we have found that our effects are not explained by population density, future work including additional control variables such as geographical region of residence will be important to confirm the robustness of these findings (Cloutier et al., 2017)." (page 31).

3. I think that the lengthy discussion of the cross-race finding (greater accuracy by Whites in decoding expressions of Black faces on the RME test) in discussing the findings of Study 2 is premature. Since the target faces in the White and Black RME tests were those of different people altogether, it could be that the Black targets for any number of different reasons were better at encoding the expressions in the first place. This interpretation is not necessarily undermined by the fact that a mentioned pilot test didn't find the same main effect (or at least I do not know enough about the pilot test to believe that the new finding is the is something worth interpreting as it is here).

We agree and have scaled back our discussion of the main effect of target race. It is no longer its own section of the discussion and is instead one paragraph on page 29 highlighting the result primarily as a future direction.

“Although it was not the primary objective of the study, the overall difference in target race accuracy in Study 2 was not predicted (note target race did not modulate any other effects). The direction of this effect, wherein White perceivers were more accurate at inferring mental states from Black than White eyes, may seem particularly surprising given past work on cross-race emotion recognition (e.g., Elfenbein & Ambady, 2002, 2003; Prado et al., 2014; Wickline et al., 2009) and that the Black RME targets were specifically equated on difficulty with the White RME targets (Handley et al., 2019). Although the current data cannot fully explain this finding, one possible explanation for this effect is that our White participants were more accurate at inferring mental states from Black targets because these targets’ emotional displays were perceived as more relevant to these participants. This possibility is consistent with previous work showing that the cross-race recognition deficit is eliminated when White people attempt to recall angry Black faces (Ackerman et al., 2006). Future research is needed to explore how racial and cultural group membership influences both primary and secondary emotion recognition.” (page 29)

Reviewer: 2

Comments to the Author(s)

The current manuscript reports two studies examining how intergroup contact influences the ability to infer others’ mental states.

The paper is well written and the literature proposed to support Authors' viewpoint is adequate. Also, statistical analyses are sound. The results are interesting and support the conclusions.

I have some points that the Authors should consider in revising their work.

Firstly, the Authors introduced their studies by summarizing intergroup contact literature with some references to neuroimaging studies. However, the evidence supporting this argument is rather limited at this point. I think it could be useful to delve a little deeper into them, by considering recent reviews such as Han, 2018 TICS, Bagnis et al. 2019 Neurosci. Biobehav. Rev, Bagnis et al. 2020 Neuroimage.

We thank you for highlighting these additional relevant citations. We have added the following sentences to the relevant paragraph on page 6 of the introduction: “Additional work suggests that contact exerts important top-down influence on social vision, particularly the neural substrates of social categorization and race perception (Bagnis et al., 2019; Bagnis et

al., 2020). Furthermore, contact may reduce the racial ingroup bias in empathy, as evidenced by a positive association between contact and anterior cingulate cortex activity when viewing outgroup members in pain (for review, see Han, 2018)."

Also, authors should specify in the introduction that they used two different versions of the RME, i.e. the original one (Baron-Cohen et al., 2001) and the Black RME (Handley et al., 2019).

We agree that this should be clearer. We have added the following sentence to page 4: "We specifically used two versions of the RME task: one version that uses exclusively White stimuli (Baron-Cohen et al., 2001) and one version that uses these same White stimuli and Black stimuli that expressed the same target emotions and were equated on difficulty with the White stimuli (see Handley et al., 2019)."

Finally, I am not sure if I follow the authors' logic of using intergroup face processing models to explain some of their results (p.28). As they state later (p.29), the RME task does not involve the processing of a whole face.

We agree that inferences from the whole face perception research may not extend directly to processing of the eye-region only. We believe that this research represents important literature to review as some studies suggest that face processing effects are mediated by the eye region (e.g., Itier, Alain, Sedore, & McIntosh, 2007; Issa & DiCarlo, 2012). However, we totally agree that one should not assume a 1-to-1 theoretical relationship between research on face processing and research with eye only and that this represents an interesting future area of research. Therefore, we have added the following sentence in the discussion to more directly address this (p. 31).

"This research primarily uses the face perception literature to derive our hypotheses as some previous research indicates that face processing effects are mediated by the eye region (e.g., Itier, Alain, Sedore, & McIntosh, 2007; Issa & DiCarlo, 2012). However, it remains unclear whether our results would generalize to situations where individuals view the whole face where they have access to additional featural cues. For example, research suggests that individuals fixate on both the mouth and eyes when processing happy faces while individuals primarily rely on the eyes for processing angry faces (e.g., Eisenbarth & Alpers, 2011). Therefore, in order to generalize these findings, future research should directly compare how contact shapes mentalizing when individuals view the eye region only versus when they view the whole face."

===PREPARING YOUR MANUSCRIPT===

- one version identifying all the changes that have been made (for instance, in coloured highlight, in bold text, or tracked changes);
- a 'clean' version of the new manuscript that incorporates the changes made, but does not highlight them. This version will be used for typesetting if your manuscript is accepted.

===PREPARING YOUR REVISION IN SCHOLARONE===

Please ensure that you include a summary of your paper at Step 2 'Type, Title, & Abstract'. This should be no more than 100 words to explain to a non-scientific audience the key findings of your research.

This will be included in a weekly highlights email circulated by the Royal Society press office to national UK, international, and scientific news outlets to promote your work.

-- Ensure that your data access statement meets the requirements at <https://royalsociety.org/journals/authors/author-guidelines/#data>. You should ensure that you cite the dataset in your reference list. If you have deposited data etc in the Dryad repository, please include both the 'For publication' link and 'For review' link at this stage.
-- If you are requesting an article processing charge waiver, you must select the relevant waiver option (if requesting a discretionary waiver, the form should have been uploaded at Step 3 'File upload' above).
-- If you have uploaded ESM files, please ensure you follow the guidance at <https://royalsociety.org/journals/authors/author-guidelines/#supplementary-material> to include a suitable title and informative caption. An example of appropriate titling and captioning may be found at https://figshare.com/articles/Table_S2_from_Is_there_a_trade-off_between_peak_performance_and_performance_breadth_across_temperatures_for_aerobic_scope_in_teleost_fishes_/3843624.

Journal Name: Royal Society Open Science

Journal Code: RSOS

Online ISSN: 2054-5703

Journal Admin Email: openscience@royalsociety.org

Journal Editor: Andrew Dunn

Journal Editor Email: openscience@royalsociety.org

MS Reference Number: RSOS-202137

Article Status: SUBMITTED

MS Dryad ID: RSOS-202137

MS Title: Impact of interracial contact on inferring mental states from facial expressions

MS Authors: Handley, Grace; Kubota, Jennifer T.; Li, Tianyi; Cloutier, Jasmin

Contact Author: Grace Handley

Contact Author Email: gandley@udel.edu, gracehandley@gmail.com

Contact Author Address 1:

Contact Author Address 2:

Contact Author Address 3:

Contact Author City: Newark

Contact Author State: Delaware

Contact Author Country: United States

Contact Author ZIP/Postal Code: 19716-5600

Keywords: mentalizing, motivation, theory of mind, intergroup contact

Abstract: Although decades of research have shown that intergroup contact critically impacts person perception and evaluation, little is known about how contact shapes the ability to infer others' mental states from facial cues (commonly referred to as mentalizing). In a pair of studies, we demonstrated that interracial contact and motivation to attend to faces jointly influence White perceivers' ability to infer mental states based on facial expressions displaying secondary emotions from both White targets alone (study 1) and White and Black targets (study 2; pre-registered). Consistent with previous work on the effect of motivation and interracial contact on other-race face memory, we found that motivation and interracial contact interacted to shape perceivers' accuracy at inferring mental states from secondary emotions. When motivated to attend to the task, high contact White perceivers were more accurate at inferring both Black and White targets' mental states; unexpectedly, the opposite was true for low contact perceivers. Importantly, target race did not interact with interracial contact, suggesting that contact is associated with general changes in mentalizing irrespective of target race. These findings expand the theoretical understanding and implications of contact for fundamental social cognition.

EndDryadContent

Date 08-Feb-2021
Sent: